# Development of Microwave Filters with Tunable Frequency and Flexibility Using Carbon Nanotube Paper

**DOI:** 10.3390/nano13182497

**Published:** 2023-09-05

**Authors:** Jih-Hsin Liu, Yao-Sheng Huang

**Affiliations:** Department of Electrical Engineering, Tunghai University, Taichung 407, Taiwan; walson117@gmail.com

**Keywords:** carbon nanotube paper, buckypaper, diode, microwave, filter

## Abstract

This study aims to exploit the distinctive properties of carbon nanotube materials, which are particularly pronounced at the microscopic scale, by deploying fabrication techniques that allow their features to be observed macroscopically. Specifically, we aim to create a semiconductor device that exhibits flexibility and the ability to modulate its electromagnetic wave absorption frequency by means of biasing. Initially, we fabricate a sheet of carbon nanotubes through a vacuum filtration process. Subsequently, phosphorus and boron elements are separately doped into the nanotube sheet, enabling it to embody the characteristics of a PN diode. Measurements indicate that, in addition to the fundamental diode’s current–voltage relationship, the device also demonstrates intriguing transmission properties under the TEM mode of electromagnetic waves. It exhibits a frequency shift of approximately 2.3125 GHz for each volt of bias change. The final result is a lightweight and flexible carbon-based semiconductor microwave filter, which can conform to curved surfaces. This feat underscores the potential of such materials for innovative and effective electromagnetic wave manipulation.

## 1. Introduction

In the field of nanoscience, carbon nanotubes (CNTs) have garnered recognition due to their multifaceted properties and potential applications [1,2,3,4,5,6,7,8]. Concurrently, CNTs have emerged as a promising technology for various electronic components, such as enhancing electronic composite materials, diodes, Field-Effect Transistors (FETs), sensors, and more. In terms of voltage-controlled capacitors or varactors, CNTs have demonstrated significant improvements over semiconductors, offering higher cut-off frequencies and lower losses [9]. Owing to their superior performance, their integration into microwave-related applications has become a focal point of study [10]. Early electromagnetic wave and nanotube interaction studies were heavily concentrated on the absorption and loss of microwaves by carbon tube-related composite materials, which subsequently expanded into electromagnetic interference (EMI) protection research [11,12,13,14,15,16,17,18].

A retrospective look at both past and recent research in this field uncovers some intriguing discoveries [19]. One noteworthy study provides a comprehensive review of scientists’ efforts to develop efficient microwave absorbers using CNTs. Deep-dive examinations [20,21,22] into the production, characteristics, and usage of CNTs in microwave absorption and millimeter-wave applications have taken place. These investigations underscored the value of CNTs within the microwave range, yet they also pointed out the need to further explore and optimize specific frequency bands and their corresponding equivalent capacitance values.

However, these studies have not entirely met the needs of CNTs in microwave applications, particularly in adjusting the absorption frequency values by modulating the external bias. Early attempts have been made to establish application models and validate variable capacitors and phase shifters for CNTs [23,24,25]. After a while, one lab proposed an example of a CNT-based microwave filter [26]. Nevertheless, the results of frequency adjustment in their measurements did not yield desirable outcomes, and the CNT area was not small and was densely populated, leading to defects during the manufacturing process.

The same experimental team subsequently improved the experiment by using carbon nanotube films, enhancing the frequency adjustability and compactness of the microwave filter circuit area [27]. Despite these promising findings, there are still some unresolved issues. For instance, the design of current CNT microwave filters is based on first fabricating a microwave filter with a metallic film layout on a high-frequency circuit board, then placing the CNT film that would exhibit minute carbon tube position deformations due to applied bias, resulting in capacitance changes, into the circuit to achieve slight adjustments near the designed electromagnetic wave absorption frequency. This entire filtering circuit has dimensions on the order of several centimeters, which is not particularly compact. On the other hand, optimizing the CNT manufacturing process and integrating them into larger mobile carriers for efficient electromagnetic protection or radar wave absorption is an important subject that needs to be addressed.

Although significant progress has been made, there remain some key experimental concepts in our understanding that need to be improved, especially regarding the absorption of electromagnetic waves in specific frequency bands using CNTs. The primary goal of this research is to fill this experimental gap and propose a new method to optimize the CNT manufacturing process for specific applications. Our research strategy is rooted in an experimental method that weaves CNTs into a flexible, wrappable semiconductor substrate similar to paper. The substrate undergoes N-type and P-type carrier doping processes to form diode element characteristics. Our method focuses on adjusting the equivalent capacitance required for different serial or parallel configurations, and we believe this could offer a better way to control the absorption of electromagnetic waves within a specific frequency range. As we continue to research and develop the manufacturing process, we are confident that our findings could lay a solid foundation for a broad range of applications, from electromagnetic protection to radar wave absorption.

## 2. Experimental Procedure

In the first phase of the fabrication process, we initially produce an independent and malleable carbon semiconductor substrate, referred to as a nanotube paper or “Buckypaper” (BP), as illustrated in Figure 1. We start by placing 60 mg of multi-walled carbon nanotube powder in a beaker, adding deionized water and Triton-x as a dispersant, and diluting it to 1% by weight. Due to the inherent Van der Waals forces attracting the nanotubes to each other, they often tangle into clumps, necessitating the use of an ultrasonic disruptor to disperse the carbon nanotube powder. We begin with 500 mL of the diluted dispersant, causing the nanotubes to disperse relatively evenly within the solution.

Subsequent steps involve diluting the solution with deionized water into ten separate 500 mL solutions containing carbon tube powder and trace amounts of dispersant. Each solution is subjected to the ultrasonic disruptor to evenly disperse the nanotubes. After dispersion, the solutions are left to settle for two hours. Ideally, due to the Brownian motion of water molecules, the nanotubes should remain suspended and not sediment. If sedimentation is observed, the ultrasonic disruptor is used continuously until the nanotubes are evenly dispersed. Once the dispersion process is complete, a vacuum filtration method is employed to retain the BP on the filter paper. The paper is then soaked in isopropanol to dissolve and wash away the Triton-x, followed by a rinse with deionized water to clean the BP. After air-drying and removing the BP, its dimensions are measured to be a circular thin sheet with a diameter of 6.5 cm and a thickness of 50 µm. Porosity checks reveal a value of 78%, implying that the nanotube powder occupies only 22% of the physical space within the buckypaper. After the fundamental carbon material substrate is fabricated, the next step is to examine the basic material properties of the nanotube paper, such as Raman spectra, to ascertain whether the crystalline structural properties maintain the characteristics of semiconducting graphite material.

Next, we attempt to turn BP into a functional semiconductor device. We begin by doping BP with both N-type and P-type conductive carriers. First, we dissolve phosphorus pentoxide powder in deionized water at a concentration of 7.5% by weight to create a phosphoric acid solution. Using thermal diffusion at 450 °C for 20 min, phosphorus elements are doped onto one side of the buckypaper, creating an N-type semiconductor. On the other side of the BP, we use a similar process. However, we thermally diffuse a 5% by weight boric acid solution, created from boron trioxide, for 20 min at 450 °C. This process increases the concentration of P-type carriers. We denote the diode created from this process as a moderately doped buckypaper diode, or M_BP diode.

Following the same procedure, but using phosphorus pentoxide and boron trioxide solutions with respective concentrations of 15% and 10% by weight, we fabricate a PN diode. We define this as a heavily doped diode, or H_BP diode. After the separate carrier doping processes, we employ a carrier concentration analyzer and employ the Hall measurement principle to determine the concentrations and resistivity of the N-type and P-type carriers, as outlined in Table 1.

We discovered that as the phosphorus doping concentration increases to 15%, the measured concentration of N-type carriers can reach up to 2.76 × 10^19^ cm^−3^. Similarly, as the boron doping concentration increases to 10%, the measured concentration of P-type carriers can reach as high as 9.71 × 10^18^ cm^−3^.

Next, we employ a JEOL JSM-6510 Scanning Electron Microscope (SEM) to confirm the surface morphology. Concurrently, we use an OXFORD Energy Dispersive X-ray (EDX) analysis to determine the percentage composition of the material’s elements, verifying the doping concentration ratio of N-type and P-type carriers.

Following the characterization of individual material properties, we proceed to measure the electrical properties of the basic components. We first use an Agilent E5270B Parameter Analyzer coupled with a dedicated two-terminal device fixture. This setup allows us to measure the basic electrical property relationship of the DC current to voltage in the nanotube paper PN diode. We also use the Agilent 4284A LCR Meter to determine the relationship between capacitance and voltage.

Finally, for microwave measurements, we employ a self-developed coaxial line microwave measurement fixture, as shown in Figure 2, to measure the S21 parameter (transmission coefficient) of the diode under different DC biases in TEM mode.

First, the sample is cut into an appropriate size of 1 cm × 1 cm, with copper foil electrodes closely applied on both sides for applying a DC bias. The sample is then clamped between a pair of Anritsu EK104F-R type coaxial adaptors. The internal diameter of the coaxial line form of this adaptor is 3.5 mm. In other words, the effective circular cross-section of the electromagnetic wave passing through the diode capacitance region has a diameter of 3.5 mm. In terms of fixture assembly, there is a polyvinyl chloride (PVC) film material, which serves as an insulator, placed between the sample and the metal part of the connector to prevent a short circuit caused by grounding the sample. Copper wires connect the central axes of the two connectors, transmitting the high-frequency signal. The area of the copper wire in contact with the sample is also coated with an insulating glue. The diode thin film acts as a dielectric, allowing electromagnetic waves to pass through it. After calibrating the HP 8722D Network Analyzer using the SOLT (short-open-load-thru) standard measurement calibration procedure, we begin to measure the S21 parameters of the electromagnetic wave spectrum between 0.05 GHz and 40 GHz passing through the carbon material diode under different DC biases in TEM mode.

Finally, using the Keysight Advance Design System Version 2015, a high-frequency circuit simulation software, we establish the high-frequency equivalent circuit model of the diode, as shown in Figure 3.

In the equivalent circuit, C_dio represents the equivalent capacitance of the diode, and Rc_dio is the series resistance at the capacitor end. These two values primarily change when a DC bias is applied to the diode. L_dio is the equivalent inductance of the diode, and Rl_dio is the series resistance at the inductor end. R_BP is the equivalent resistance of BP for electromagnetic wave background absorption. C_coaxial and R_coaxial are the parasitic capacitance and resistance generated due to induction with the grounded outer ring of the coaxial transmission line.

After establishing the equivalent circuit, we use numerical fitting methods to extract the variable capacitance values, internal impedances, and resonance absorption frequencies corresponding to the resonance absorption peaks of the high-frequency signals for moderately doped diodes, heavily doped diodes, and series diodes under different DC biases. We then observe these changes and investigate the physical reasons behind the trends.

## 3. Results and Discussion

At the initial stage of the experiment, buckypaper without carrier doping was prepared by vacuum filtration, and its Raman spectrum is shown in Figure 4. The value of the D-band is slightly higher than that of the G-band, indicating that there are quite a few defects in the arrangement of carbon atoms inside the carbon nanotubes. This can lead to higher internal impedance when the current flows through the material. In addition, the wider peak of the 2D band reveals the presence of multiple graphene layers in the component structure, which is characteristic of multi-walled carbon nanotubes. The small peaks behind the 2D band indicate a slight oxidation of the carbon nanotubes. In order to make a carbon material semiconductor diode, we separately doped the two sides of the buckypaper with boron and phosphorus elements. In addition to using a carrier concentration analyzer to detect their concentration and resistivity, we also assisted with electron microscopy, as shown in Figure 5a–d. Further, using a higher magnification of the electron microscope to observe the sample, as shown in Figure 5c, we can find that the surface morphology of the sample presents a filamentous weave, with many gaps in between the filaments, resembling cotton. This is why it looks like a piece of paper on a macroscopic scale and has a high degree of flexibility. After performing elemental analysis on each sample with EDS, since the background material is a carbon nanotube, the carbon content of each sample generally remains around 90%. However, for the sample heavily doped with phosphorus, as shown in Figure 5d, the carbon content is only 55.07% and the oxygen content is 32.57%. The suspected reason is that after a high concentration of heavily doped solution is evaporated and dried at a high temperature, besides phosphorus elements penetrating into the carbon structure, a large number of crystalline particles of phosphorus oxide will also appear. The EDS measurement area framed a lot of oxide crystalline particle areas, so the proportion of oxygen elements is higher. Overall, the sample heavily doped with boron elements, as measured by EDS, has a boron content weight percentage of 6.25%, which is higher than the 4.25% of moderately doped ones. The weight percentage of phosphorus content in the heavily doped phosphorus sample is 12.36%, which is also higher than the 5.19% of moderately doped ones. All samples showed effective P-type and N-type carrier doping, and generally, heavy doping shows a more concentrated ratio and moderate doping shows a lighter ratio relationship.

After making the carbon material diode, we began to measure the basic electrical characteristics of the components. From Figure 6a, we can see that for a single moderately doped diode, the current rises sharply after a forward bias of about 0.4 volts. When two diode components are stacked together to act as a series component, it is shown that the DC voltage is approximately 0.7 volts before the current conducts.

The same situation also occurs in heavily doped diode components, such as in Figure 6b. The conduction voltage of the series components is about 1.0 volts, which is more than twice the conduction voltage of a single component of 0.3 volts. The suspected reason is that when stacking layers to form series components, there may be oxide crystalline particles in the gap between the overlapping of the two components, which directly increases the value of the contact resistance. As a result, the series components need to apply a conduction voltage that is more than twice that of the diodes to make the current conduct.

Further, we use the equation of current and voltage for a non-ideal diode as follows:(1)$$I=I_{S}\cdot (e^{\frac{qV}{nKT}}-1)$$

In the equation, *V* is the bias applied to the diode; *I* is the current generated after the diode is biased; *I_s_* is the reverse saturation current; *n* is the ideal factor; *q* is the Coulomb charge; *K* is the Boltzmann constant; *T* is the absolute temperature.

By substituting the various constants and the room temperature T = 300 K to perform data fitting, and extracting the value of the reverse saturation current, namely *I_s_*, and the ideal factor, namely n. Generally speaking, the n values of all the components lie within the range of 1 to 10, substantially greater than the typical 1 to 2 range for silicon-based diodes. In other words, at the same bias, the speed of current rise is not fast enough, that is, the internal impedance is larger. It is speculated that in BP, the conduction of electrons relies on the contact between nanotubes, and then electrons can move to generate current, but countless carbon tubes also contribute countless contact resistances.

In a single moderately doped diode, *I_s_* is 1.3 × 10^−3^ Ampere, and the leakage current is slightly larger. It is speculated that on the one hand, the doping concentration is not high enough, so the formed PN junction appears not robust enough. On the other hand, BP is interwoven with carbon tubes and air. When doping carriers, it is impossible to form a whole PN junction like a solid silicon-based semiconductor. A small number of unevenly doped carbon tubes are likely to cause leakage current. In comparison, after significantly increasing the doping concentration of phosphorus and boron, the heavily doped diodes made have an n value of 1.8 and an *I_s_* value of 4.0 × 10^−5^ Ampere, and their electrical characteristics have greatly improved compared to moderately doped diodes.

Beyond comprehending the relationship between current and voltage in semiconductor components, it is equally pivotal to investigate the relationship between diode variable capacitance and direct current bias. As depicted in Figure 7a,b, when a forward bias is applied, the width of the diode’s depletion region narrows continuously in accordance with the basic physics of the PN junction, leading to an increased capacitance value in the depletion layer as the forward bias rises. Concurrently, in the case of series-connected diode components, due to the bias being shared evenly among all elements, a more significant bias is required to induce a notable alteration in the series capacitance value.

Returning to our primary focus, the penetration and absorption of high-frequency electromagnetic waves by carbon-based diode components, we utilize a network analyzer. We emit electromagnetic waves from port 1, which penetrate through the diode component and are received at port 2. We observe the microwave spectrum response under different direct current biases for diode components manufactured under varying processing conditions, as shown in Figure 8a–d.

We are aware that undoped buckypaper inherently possesses electromagnetic wave loss characteristics. At a frequency of 10 GHz, there is a loss of approximately −1 dB, which incrementally increases with rising frequency [28]. When undoped buckypaper undergoes a doping process to form a diode, the addition of a PN junction’s depletion region capacitance makes it feasible to design and trap specific microwave frequency bands within the diode for absorption. Subsequently, when we alter the direct current bias, the capacitance value of the diode’s depletion region progressively rises with the application of a forward bias, resulting in an increased capacitance value.

In general electronics, the resonant frequency response formula for a parallel circuit of capacitance and inductance is given by:(2)$$f_{resonate}=\frac{1}{2\pi \sqrt{L\cdot C}}$$

In which *f_resonate_* represents the resonant frequency of a parallel circuit composed of capacitance and inductance; *L* denotes the inductance; and *C* signifies the capacitance.

From the aforementioned equation, it can be predicted that as the capacitance increases, the resonant frequency will decrease. Further observations from Figure 8a–d confirm this, as when the forward bias increases, the resonant frequency of each diode device moves to a lower frequency to varying degrees.

Moreover, regardless of whether a diode is moderately doped or heavily doped, when two diodes are connected in series, the equivalent capacitance, in accordance with basic physics, will be smaller than that of a single diode, leading to an increased resonant absorption frequency. Thus, the resonant frequency of the series-connected moderately doped diode in Figure 8b is higher than that of the single diode in Figure 8a. Similarly, the resonant frequency of the series-connected heavily doped diode in Figure 8d is higher than that of the single diode in Figure 8c.

We attempted to extract the spectrum response data from Figure 8c, representing the heavily doped diode at a 0 volt bias, and employed the equivalent circuit from Figure 3 to execute a high-frequency equivalent circuit simulation. This process allowed us to determine the equivalent values of each high-frequency resistor, capacitor, and inductor in the circuit. We had previously employed an LCR meter to measure the inductance and voltage relationship of the buckypaper diode. The results essentially remained constant. Thus, in our circuit simulation software, “Keysight Advance Design System Version 2015”, we also defined the inductance as invariant with voltage changes. The simulation results closely overlap the measured data, as shown in Figure 8e. By slightly adjusting the equivalent diode capacitance value (C_dio) and the series resistance value (R_dio), we could successfully overlap the data corresponding to the resonant absorption peaks of each diode device in Figure 8a–d. The relationship between the extracted C_dio and the applied bias is illustrated in Figure 9a, and the relationship between the extracted R_dio and the applied bias is shown in Figure 9b. For instance, observing the transmission spectrum response in Figure 8d for the M_BP_diode_two_in_series, we notice that as the forward bias increases, the peak of resonant absorption shifts towards a lower frequency, implying that the equivalent capacitance also increases with the rising bias. This is further corroborated by Figure 9a, which shows an increasing trend of the extracted capacitance value with the escalating bias for M_BP_diode_two_in_series. On the other hand, in Figure 8d, as the forward bias increases, the transmission loss of the electromagnetic wave also decreases. This implies that the equivalent high-frequency resistance value decreases with the rising bias, a trend that is also confirmed by Figure 9b.

Finally, returning to our interest in the range of electromagnetic wave absorption frequencies that can be adjusted by bias, we define the resonant frequency at a 0 V bias as the origin frequency. As the forward bias increases, the frequency of electromagnetic wave absorption shifts to an upper frequency. Subsequent resonant frequencies minus the origin frequency serve as the offset frequencies, as illustrated by the relationship with the applied bias in Figure 9c. Experimental data show that the adjustable frequency range of a single or series-connected heavily doped diode is greater than that of a moderately doped diode. Among them, the H_BP_diode_single, when the bias ranges from 0.1 V to 0.4 V, displays the broadest offset of resonant absorption frequency from 0.2 GHz to 0.925 GHz. This translates to an adjustment rate of 2.3125 GHz/Volt, which can be deemed as reaching an applicable range in the field of electromagnetic wave communication.

## 4. Conclusions

In this experiment, we attempted to use a very light carbon material semiconductor, that is, a carbon nanotube. We interwove it to form a semiconductor substrate resembling paper, possessing flexibility and wrap-ability. After the carrier doping process was introduced to form the characteristics of diode elements, this optimized component was able to selectively absorb high-frequency electromagnetic waves in response to changes in DC bias, and it allowed for the adjustment of the frequency range.

The carbon material diodes developed in our experiments are semiconductor active devices. They allow real-time adjustments of different absorption losses and frequencies simply by tuning the applied voltage. If it can be laid out over a large area on the surface of certain mobile carriers, it can be endowed with functions such as electromagnetic protection or radar wave absorption. However, to achieve the absorption of electromagnetic waves in certain specific frequency bands, one must first design the equivalent capacitance corresponding to the required frequency in different series or parallel configurations, so that the absorption of electromagnetic waves can be adjusted around a specific frequency range. Achieving this will require engineers to continue to work hard on the research and development of different manufacturing processes.

## Figures and Tables

**Figure 1 nanomaterials-13-02497-f001:**
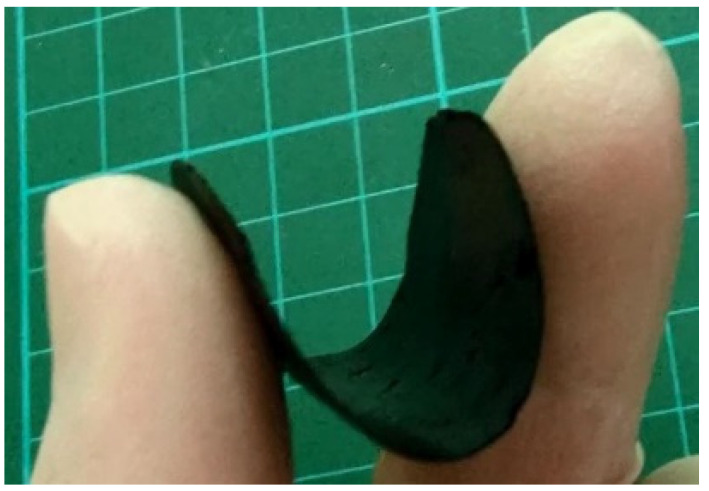
Appearance of flexible carbon nanotube paper.

**Figure 2 nanomaterials-13-02497-f002:**
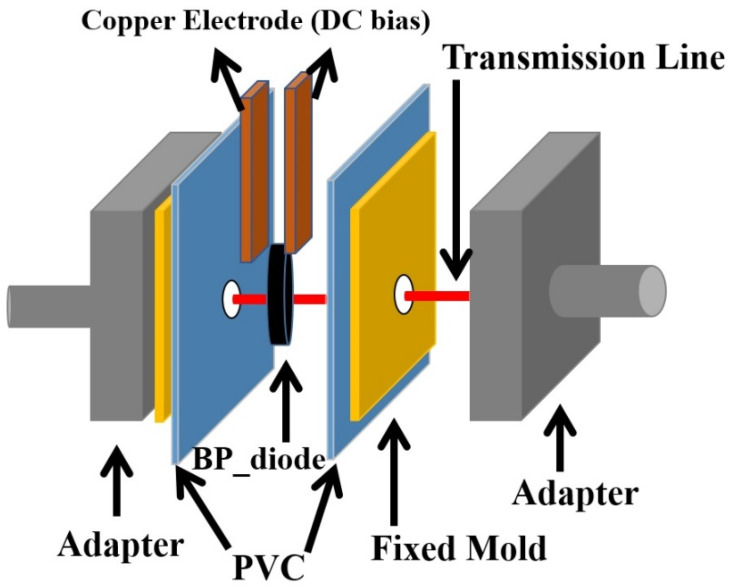
Measurement setup for electromagnetic characteristics of carbon-based diode.

**Figure 3 nanomaterials-13-02497-f003:**
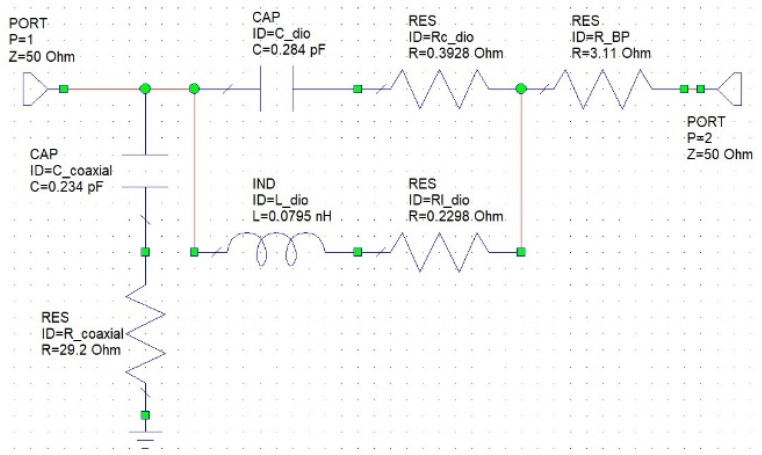
The high-frequency equivalent circuit of this carbon material diode was simulated using the “Keysight Advance Design System Version 2015” high-frequency circuit simulation software, with a frequency range spanning from 0.05 GHz to 40 GHz.

**Figure 4 nanomaterials-13-02497-f004:**
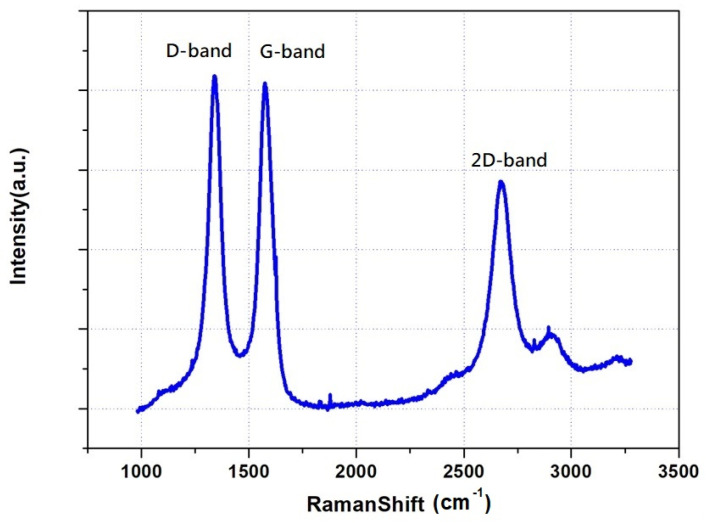
Raman spectrum of nanotube paper.

**Figure 5 nanomaterials-13-02497-f005:**
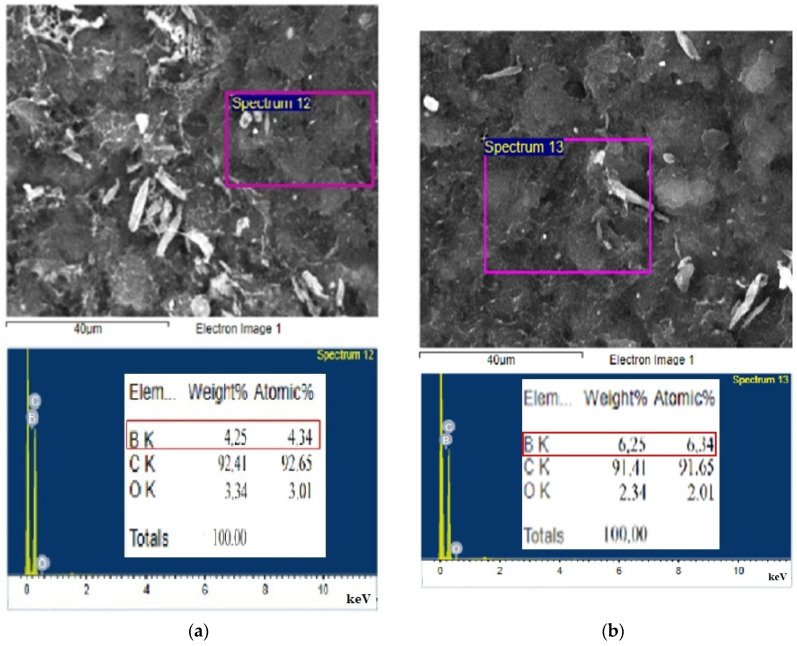
SEM morphology and EDS analysis of doped nanotube paper: (**a**) 5% boron-doped; (**b**) 10% boron-doped; (**c**) 7.5% phosphorus-doped; (**d**) 15% phosphorus-doped.

**Figure 6 nanomaterials-13-02497-f006:**
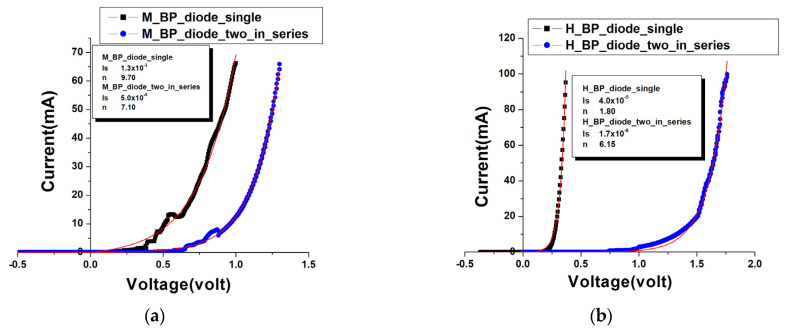
Current-voltage characteristic of BP diodes: (**a**) single and series-connected moderately doped diodes; (**b**) single and series-connected heavily doped diodes.

**Figure 7 nanomaterials-13-02497-f007:**
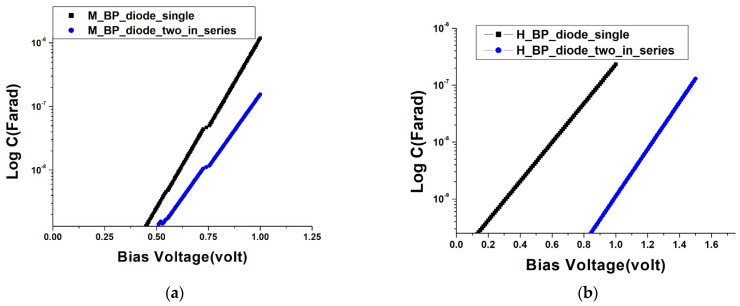
Capacitance-voltage characteristics of BP diodes: (**a**) single and series-connected moderately doped diodes; (**b**) single and series-connected heavily doped diodes.

**Figure 8 nanomaterials-13-02497-f008:**
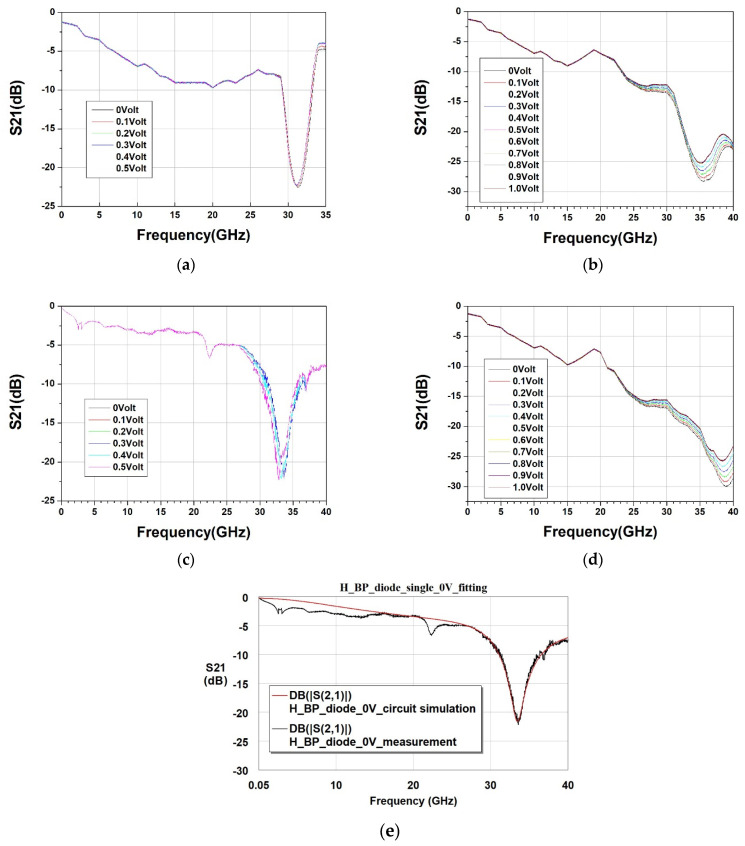
S21 parameter spectra of BP diodes under different DC biases: (**a**) single moderately doped diode; (**b**) two series-connected moderately doped diodes; (**c**) single heavily doped diode; (**d**) two series-connected heavily doped diodes; (**e**) measurement data and high-frequency equivalent circuit simulation for a single heavily doped diode at 0 Volt DC bias.

**Figure 9 nanomaterials-13-02497-f009:**
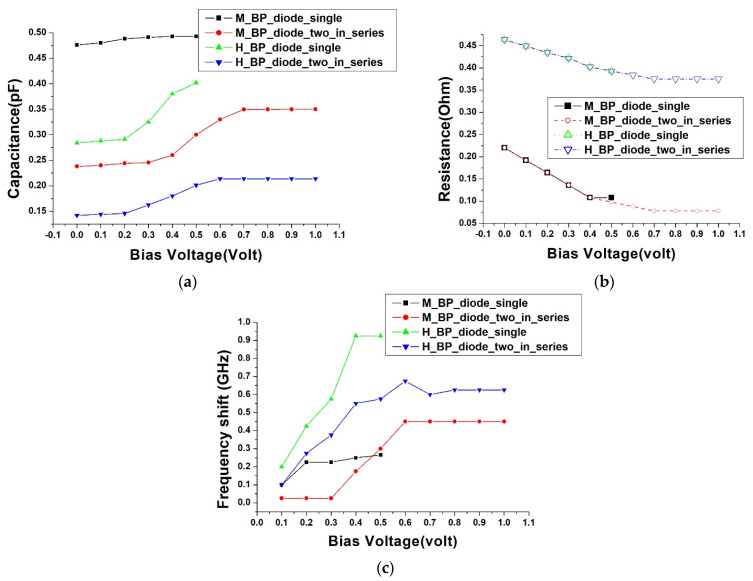
Relationships between extracted equivalent circuit values from S21 spectra and bias voltage: (**a**) equivalent diode capacitance C_dio; (**b**) equivalent diode impedance R_dio; (**c**) offset of equivalent resonant frequency.

**Table 1 nanomaterials-13-02497-t001:** The resistivity, carrier concentration, and semiconductor polarity of BP under different doping conditions are pivotal properties that directly influence its electronic behavior.

Doping Concentration	Bulk Resistivity [Ohm-cm]	Bulk Carrier Density [cm^−3^]	Type
Un-doped BP	3.15 × 10^−2^	1.92 × 10^16^	P Type
Boron 5%	3.02 × 10^−2^	1.32 × 10^17^	P Type
Boron 10%	3.55 × 10^−2^	9.71 × 10^18^	P Type
Phosphorus 7.5%	3.35 × 10^−2^	1.15 × 10^17^	N Type
Phosphorus 15%	3.42 × 10^−2^	2.76 × 10^19^	N Type

## Data Availability

The data presented in this study are available on request from the corresponding author.

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
