# Peer review of "Development of Microwave Filters with Tunable Frequency and Flexibility Using Carbon Nanotube Paper"

_nanomaterials, 2023, doi:10.3390/nano13182497_

Round 1
Reviewer 1 Report
Report on "Development of Microwave Filters with Tunable Frequency and Flexibility using Carbon Nanotube Paper." This manuscript reports on the use of a carbon nanotube based semiconducting device to modulate EM wave absorption. I found the results to be interesting and relevant to Nanomaterials. Therefore, I recommend it for publication following some issues about the material itself.
1. The authors discuss the fabrication process using MWCNTs. Why did the authors choose MWCNTs which are inherently more defective, less flexible, and will provide less connectivity to the BP material than SWCNTs? Electrical conductivity is one important factor for this material performance as the authors indicated in the text. Please comment. Also, please comment on the expected difference using SWCNTs, which would result in higher packing, smaller pores, and higher densities.
2. In the SEM Images, what are all of the large white features (particles)? They are excluded from the EDX examination, but are in large population.
3. How do these results compare with other attempts at EM absorption? This would assist in clarifying the uniqueness of these approach and results.
Reviewer 2 Report
By impregnating both faces of a C nanotube buckypaper by phosphorus- and boron- containing solutions, the authors have realized NP-like diode films. Two families of samples were realized, moderately-doped and heavily-doped. The electrical characteristics of the diodes, either single or associated in series of two, were characterized under DC bias. Finally, the transmission of GHz electromagnetic waves across the same devices were measured as a function of the applied DC voltage.
The results are generally interesting. However, the paper must be improved in several ways.
The observed resonance frequency follows eq. (2) (line 280) only qualitatively if one compares Fig. 9(c) and Fig. 9(a). For instance, the red curves indicate a sudden jump of the capacitance between 0.5 and 0.6 V, whereas the resonance frequency varies more smoothly between 0.3 and 0.6 V. Presumably, the inductance also varies with the bias. Do the authors have information on that?
The authors insist on the potential applications of their devices as a microwave filter with tunable frequency. According to Fig. 8(e), the absorption measured from the background transmission loss can reach 15 dB at the resonance frequency. Is this attenuation sufficient in practice? More importantly, perhaps, what is the reproducibility of the diode properties from one batch to another? Have the authors investigated that point? What is the nature of the crystalline particles mentioned in line 194 for the phosphorus H-BP samples?
Figs. 1, 2 and 5 apart, the quality of the figures is not good enough and the lettering used for the drawing is too small. Fig. 4 is acceptable, except for the label of the Raman D line that is written as "D-bnad" and must be corrected. The authors must indicate for what device and what frequency are the values of the elements of the equivalent circuit they have indicated in Fig. 3.
There are a few typographical errors in the text that need correction:
* Line 317: "lower frequency" should be "upper frequency"
* Line 321: "M_BP_diode_single" should be "H_BP_diode_single".
The English is ok, the sentences are understandable without amiguity
Reviewer 3 Report
This paper demonstrates the applications of microwave filters using carbon nanotubes paper. I consider that this paper shows interesting possible applications using a facile technique with fascinating properties of carbon nanotubes. My comments are listed below.
1) How to prepare carbon nanotubes?
2) I recommend showing Raman spectra after boron and phosphorus doping into carbon nanotubes.
3)What is the TEM mode in this paper?
4) There are no units in the horizontal lines in Fig. 4.
Round 2
Reviewer 2 Report
I am generally happy with the responses of the authors, the corrections performed in the text and the modifications of the figures.
I would have been more happy if the authors have added at least a few elements of their responses in their revised manuscript. For instance, it seems important to tell the readers that, according to electrical measurements they have performed on the diodes, the authors have considered the inductance as invariant with voltage changes in the circuit simulations.
Moreover, the sentence "The carbon material diodes developed in our experiments are semiconductor active devices. They allow real-time adjustments of different absorption losses and frequencies simply by tuning the applied voltage" is an important point to be underlined in the conclusions.
I warmly suggest the authors comply with these very minor remarks, which I believe will contribute to improve further the quality of their paper.
Reviewer 3 Report
The authors replied and revised the manuscript appropriately to the referees's comments. Therefore, I recommend the publication of this paper.
